# Predicting Modifiers of Genotype-Phenotype Correlations in Craniofacial Development

**DOI:** 10.3390/ijms24021222

**Published:** 2023-01-08

**Authors:** Ranjeet D. Kar, Johann K. Eberhart

**Affiliations:** Department of Molecular Biosciences, College of Natural Sciences, University of Texas at Austin, Austin, TX 78712, USA

**Keywords:** phenotypic variation, zebrafish, craniofacial development, neural crest, Gata3, transcriptomics

## Abstract

Most human birth defects are phenotypically variable even when they share a common genetic basis. Our understanding of the mechanisms of this variation is limited, but they are thought to be due to complex gene-environment interactions. Loss of the transcription factor Gata3 associates with the highly variable human birth defects HDR syndrome and microsomia, and can lead to disruption of the neural crest-derived facial skeleton. We have demonstrated that zebrafish *gata3* mutants model the variability seen in humans, with genetic background and candidate pathways modifying the resulting phenotype. In this study, we sought to use an unbiased bioinformatic approach to identify environmental modifiers of *gata3* mutant craniofacial phenotypes. The LINCs L1000 dataset identifies chemicals that generate differential gene expression that either positively or negatively correlates with an input gene list. These chemicals are predicted to worsen or lessen the mutant phenotype, respectively. We performed RNA-seq on neural crest cells isolated from zebrafish across control, Gata3 loss-of-function, and Gata3 rescue groups. Differential expression analyses revealed 551 potential targets of *gata3*. We queried the LINCs database with the 100 most upregulated and 100 most downregulated genes. We tested the top eight available chemicals predicted to worsen the mutant phenotype and the top eight predicted to lessen the phenotype. Of these, we found that vinblastine, a microtubule inhibitor, and clofibric acid, a PPAR-alpha agonist, did indeed worsen the *gata3* phenotype. The Topoisomerase II and RNA-pol II inhibitors daunorubicin and triptolide, respectively, lessened the phenotype. GO analysis identified Wnt signaling and RNA polymerase function as being enriched in our RNA-seq data, consistent with the mechanism of action of some of the chemicals. Our study illustrates multiple potential pathways for Gata3 function, and demonstrates a systematic, unbiased process to identify modifiers of genotype-phenotype correlations.

## 1. Introduction

Congenital birth defects are among the leading causes of infant mortality in the United States, according to the Centers for Disease Control. Defects of the craniofacial skeleton are among the most common birth defects. For example, the prevalence of cleft lip with or without cleft palate is 10.2 per 10,000 births in the United States [1]. In addition, phenotypic variation is common within birth defects. The causes of these defects and their variability is not well understood, but is thought to result from a combination of genetic and environmental factors.

One major neural crest subpopulation, cranial neural crest cells (CNCCs), is the primary source of progenitors of skeletal and connective tissues in the face [2]. In both humans and zebrafish, CNCCs that generate the palatal skeleton populate the maxillary prominence of the first pharyngeal arch, as well as the frontonasal prominences [3,4,5]. In zebrafish embryos, the cartilaginous palate (anterior neurocranium) consists of an ethmoid plate connected posteriorly to bilateral trabeculae, and forms the roof of the oropharynx [4,5]. The trabeculae serve as the connection to the posterior neurocranium, which is mostly derived from mesoderm [6]. Functional analysis has shown that early zebrafish development, including palatal growth, can be used to model craniofacial development in mammals [7,8,9]. Due to their external development and genetic tractability, this makes zebrafish an excellent model in which to identify and characterize gene-environment interactions.

The zinc finger transcription factor Gata3 is one in a family of factors that regulate cell behaviors related to development, including differentiation, migration, and motility [10,11,12]. *GATA3* haploinsufficiency causes hypoparathyroidism, sensorineural deafness and renal dysplasia (HDR) syndrome in humans, which can include palate defects [13,14]. Most notably, *GATA3* has been associated with microsomia in humans [15]. Loss of function experiments in mice show that *Gata3* is necessary for the proper development of structures derived from cranial neural crest cells, including the tongue, mandible and teeth [16,17]. As a transcription factor, Gata3 mediates the interactions of multiple cofactors and chromatin to control gene expression. Despite the link between Gata3 and palate formation, the genetic mechanisms by which it regulates craniofacial development are largely unknown.

Previous work by our lab shows that a point mutation in zebrafish *gata3* results in palate defects and recapitulates the phenotypes observed in human HDR patients [18]. Phenotypes associated with GATA3 in humans are highly variable. While HDR is quite rare, microsomia is more common, with an incidence rate of at least 1 in every 5600 embryos [19]. We have shown that the craniofacial phenotypes in *gata3* mutant zebrafish are also highly variable and depend upon genetic background [18,20]. The variability in both human and zebrafish *GATA3* mutants are likely due to variable disruption of the GATA3 transcriptional network, possibly due to environmental perturbation. Here, we use bioinformatic approaches to predict chemical modifiers of the *gata3* mutant phenotype. Our results demonstrate that the utility of the approach and suggest that similar approaches could be used to identify modifiers of other mutations.

## 2. Results

### 2.1. Targets of Gata3 in Cranial Neural Crest Cells

Previous work in zebrafish demonstrated that an intersectional approach, collecting cells positive for both *fli1:EGFP* and *sox10:mRFP* transgenes via FACS, could be used to collect a pure population of cranial neural crest cells (CNCC) [21]. We used this approach to collect CNCC from 28 hpf zebrafish embryos, a time point when Gata3 function is critical for proper craniofacial development [18].

To determine the effect of *gata3* on craniofacial development, we collected CNCCs from control, knockdown and rescue groups from three biological replicates and conducted RNA-seq (Figure 1). The control group comprised embryos that were not injected with a *gata3* morpholino, the knockdown group was injected with the morpholino, and the rescue group was injected with the morpholino and subsequently experienced ectopic *gata3* expression due to a heat-shock-driven transgene. The resulting RNA-seq dataset yielded 701 differentially expressed genes, comparing each treatment to control. Legitimate targets of *gata3* are predicted to be those whose expression moved in opposite directions when comparing the knockdown and rescue datasets (Figure 2A). This final group comprises 551 genes (Figure 2B, false-discovery rate < 0.05). Gene Ontology (GO) term enrichment [22] indicated that at 5 of the top 6 most-enriched terms were those related to either the Wnt pathway or RNA polymerase II activity (Figure 3). More specifically, we found enrichment of terms tied to RNA polymerase II activity, driven by DNA binding. The top three upregulated and downregulated genes are shown in Figure 3B. Our GO analysis also revealed links to Wnt protein binding, with the top three upregulated and downregulated genes shown in Figure 3C. To ensure that the sample was not contaminated by non-CNCCs, we looked at the expression values of *flk1* (vasculature) and *otx2* (otic placodes). Neither of these genes had high read counts in our dataset, allowing us to conclude that our data are highly enriched in legitimate targets of *gata3* in CNCC.

To validate our RNA-seq data, we selected the ten genes (five upregulated, five downregulated) with the highest magnitude of expression change (Figure 2C,D) and performed RT-qPCR. Nine of these ten genes demonstrated significant differential expression mirroring that of our RNA-seq data (Appendix A). We conclude that our RNA-seq dataset is highly enriched in Gata3 target genes within the neural crest. Principal component analysis (Appendix A) demonstrated no major batch effects. It also showed that the neural CNCC transcriptome in uninjected controls resembled that of the rescue group (with ectopic Gata3 expression), whereas the knockdown group was transcriptomically distinct, as we would expect.

### 2.2. Transcriptomic Data Can Predict Chemical Modifiers of Genotype-Phenotype Correlations

We sought to determine the extent to which our transcriptomic data could predict chemical modifiers of the mutant phenotype. From our *gata3* transcriptomic dataset, we queried the LINCS L1000 platform (https://clue.io/, accessed on 21 November 2022) [23] with the 100 top positively and negatively regulated genes (200 total genes). Our data comprise both loss- and gain-of-function approaches. For clarity, we relate these chemicals to direction of expression change in gata3 loss-of-function neural crest cells. Thus, positive correlation predicts worsening of the phenotype and inverse correlation predicts lessening. Of the chemicals predicted by LINCs, we chose the 8 most positively and 8 most negatively correlated chemicals (Figure 4) to test their ability to modify *gata3* mutant phenotypes.

We first performed a dose response analysis for each of the chemicals to identify the lowest dose with which we observe facial defects. We tested doses at the concentrations of 0.1 μM, 1 μM, 10 μM, and 100 μM. We selected the highest concentration that yielded an embryo death rate or induced gross morphological defects in less than 10% of fish (Appendix A).

Previous work in our lab has shown that *gata3^b1075^* mutant phenotypes in zebrafish are variable and established two inbred lines that consistently gave phenotypes at each end of this spectrum [18]. In one line, the trabeculae are intact but chondrocyte arrangement is altered resulting in a defect in extension of the trabeculae (referred to as the mild phenotype). In the other, the trabeculae are broken resulting in a failure in connection of the anterior and posterior regions of the neurocranium (the severe phenotype). We leveraged this variability to test for specific interactions with our candidate small molecule drugs.

Chemicals with positive correlations to the *gata3* loss-of-function data are anticipated to worsen the mutant phenotype. Therefore, we tested these chemicals on the mild phenotype background. We scored fish by counting the number of uninterrupted trabeculae in each embryo. To assay the effect of the chemicals predicted to worsen phenotypes, we determined appropriate dosing concentrations for each drug through graded assays in wild-type fish. We then used these concentrations to treat *gata3^b1075^* “mild” embryos from 24 to 48 hpf, assessing phenotype in embryos fixed and stained with Alcian blue and Alizarin red.

Of the eight chemicals tested, two exacerbated the craniofacial phenotypes in mild *gata3* mutants. Treatment with vinblastine, a microtubule inhibitor, leads to a significant decrease in number of intact trabeculae (Figure 5F, Fisher’s exact test, *p*-value < 0.05). While the majority of untreated mutants had two intact trabeculae (14 out of a total 26), the opposite is true for exposed mutants (6/23). We took linear measures of the length and width of the trabeculae to quantify the effects on extension. Consistent with observations in *gata3^b1075^* mutant fish, we saw shorter (control = 1.0709 ± 0.02093 μm, treatment = 0.9615 ± 0.0320 μm, *p* = 0.0135) and wider (control = 0.2701 ± 0.0135 μm, treatment = 0.3313 ± 0.0137 μm, *p* = 0.00197) trabeculae in vinblastine-treated fish (Figure 5G,H).

Clofibric acid, a peroxisome proliferator-activated receptor (PPAR) agonist produced a similar result. There were two intact trabeculae in 18/23 untreated embryos and 13/29 in treated embryos. This difference was statistically significant (Figure 6G, Fisher’s exact test, *p* < 0.05). In some embryos, the small molecule treatment would result in discontinuous trabeculae formation (Figure 6E), and in others, treatment resulted in full truncation of the trabecula (Figure 6F, *p* < 0.05), similar to the *gata3^b1075^* “severe” phenotype. As with vinblastine, we observed shorter (control = 1.8703 ± 0.0595 μm, treatment = 1.6330 ± 0.0714 μm, *p* = 0.0122) and wider (control = 0.2294 ± 0.0100 μm, treatment = 0.2784 ± 0.0128 μm, *p* = 0.00333) trabeculae in clofibric acid-treated fish (Figure 6H,I).

While exacerbation of mutant phenotypes is informative, we were particularly interested in knowing if LINCS could predict chemicals that would lessen the severity of mutant phenotypes. We identified two chemicals that significantly improved phenotypes in *gata3* mutants. Treatment with triptolide, a diterpene epoxide with anti-inflammatory and antitumor properties, led to a significant change in the distribution of intact trabeculae count in embryos (Figure 7G, *p* < 0.001), with a higher proportion of treated embryos with two intact trabeculae compared to untreated embryos (15/22 and 5/25, respectively). Although phenotypes were variable, in rare cases we observed a rescue resembling wild-type embryos (Figure 7D). Our trabecular linear measurements in triptolide-treated embryos revealed embryos that more resembled wild-type fish in length (control = 1.4665 ± 0.0869 μm, treatment = 1.8579 ± 0.0803 μm, *p* = 0.0134) and width (control = 0.3172 ± 0.0157 μm, treatment = 0.2436 ± 0.0131 μm, *p* = 0.000508) than untreated *gata3^b1075^* mild fish (Figure 7H,I).

The second chemical that improved phenotypes is Daunorubicin, a chemotherapy drug used for certain leukemias. This chemical also produced a significantly higher proportion of embryos with two intact trabeculae in treated embryos (15/22) versus untreated embryos (1/18) (Figure 8D–F, Fisher’s exact test, *p* < 0.05). As expected, these embryos had longer (control = 1.2186 ± 0.1074 μm, treatment = 1.7814 ± 0.0887 μm, *p* = 0.000166) and wider (control = 0.3498 ± 0.0202 μm, treatment = 0.2340 ± 0.0143 μm, *p* = 0.0000225) trabeculae compared to untreated *gata3^b1075^* severe mutant fish (Figure 8H,I).

To ensure that our results were truly a reflection of LINCS prediction, we randomly selected 10 chemicals from the full LINCS list, that were not correlated with the input data. We treated *gata3* mutants with them as described above (Appendix A). We found no significant difference in intact trabeculae between treated and untreated groups in this sample. This outcome demonstrates that the LINCS dataset can effectively predict chemicals that will modify mutant phenotypes.

## 3. Discussion

Most birth defects are thought to be caused by complex gene-environment interactions. However, identifying environmental exposures that modify phenotypic outcomes of a genetic mutation is extremely difficult. We have demonstrated that transcriptomic profiling can be successfully used to identify chemicals that will modify this genotype-phenotype relationship in predictable ways. Our findings have translational implications as many of the drugs and chemicals in the LINCS dataset have clinical use or are found in the environment.

### 3.1. Prediction of Chemicals That Worsen Mutant Phenotypes

Our approach holds excellent promise for identifying therapies that should be contraindicated in specific populations of individuals. Vinblastine is an anticancer drug that exacerbated defects in *gata3* mutant embryos. This has notable clinical implications as vinblastine is recommended for treatment (in combination with other therapies) of pregnant patients with non-epithelial ovarian cancer [24]. Human studies have not identified rates of birth defects above what would be expected in women undergoing treatment. However, these studies are too small to detect sensitization to specific genotypes, but our findings raise the question of whether vinblastine may have deleterious effects for developing progeny of cancer patients—or whether specific drug combinations may mitigate those complications.

The PPAR agonist clofibric acid (clofibrate is its anionic form) plays a major role in fatty acid metabolism, and most notably has been demonstrated to lower triacylglycerol in rat fetuses through clofibrate administration in the maternal diet while increasing mRNA concentrations of proto-oncogenes c-myc and c-jun in the liver of pigs [25,26]. Although the hepatic effects of clofibrate in both mammalian fetuses and adults have been studied, and *gata3* knockout in embryonic mice elicits a significant phenotype in the liver the genetic interaction between the drug and the gene in the zebrafish palate is novel, and suggests that clofibrate may have more wide-ranging developmental downstream effects than previously suspected [27].

Beyond the four chemicals we independently validated to interact with *gata3* in the embryonic zebrafish palate, the LINCS dataset revealed more avenues of study to investigate possible mechanisms of action for craniofacial development. Half of the top 10 chemicals predicted to enhance the *gata3* mutant phenotype have tubulin inhibitor function. Previous chemical and genetic studies have shown that tubulin stability is regulated in part through a Wnt-activating pathway, tying together the findings from our differential expression and gene ontology analyses [28,29]. Interestingly, vinblastine and its sister derivative chemical vincristine were both identified as potential chemical modifiers with microtubule inhibitory activity via the LINCs analysis, but only vinblastine was observed to have a significant effect on the *gata3* phenotype. Comparative studies have shown that the two chemicals have similar potency but differential inhibition of net addition of tubulin in vitro [30]. That such a narrow difference in function could have a quantifiable effect on interaction with the *gata3* phenotype illustrates the potential specificity of microtubule function within this craniofacial signaling pathway.

### 3.2. Identification of Chemicals That Are Protective against Deleterious Phenotypes

While avoiding deleterious exposures may be of primary concern, understanding chemicals that protect against deleterious phenotypes could be quite useful in preventing birth defects. One of the small molecules that lessened the *gata3* phenotype was triptolide, an RNA polymerase inhibitor. Used in traditional Chinese medicine for many years for its anti-inflammatory properties, it has been more recently shown to have potential anti-cancer applications [31,32,33]. To date, its therapeutic properties have been studied with respect to transcriptional regulation of oncogenes and cell cycle regulators [34], but our work suggests that further study of triptolide’s potential protective effects in craniofacial development is warranted. Interestingly, we observed a near-complete rescue of the mutant phenotype in some instances. This raises the that triptolide’s activity in the neural crest closely resembles that of *gata3*.

Another drug that our studies showed to mitigate *gata3*-related palatal defects, daunorubicin, has also been used in anticancer combination therapies for several decades, including one case in which it contributed to complete remission of leukemia in a pregnant woman [35]. However, daunorubicin also elicits cytotoxicity by potentially inhibiting DNA repair pathways [36]. In zebrafish embryos, anthracyclines such as daunorubicin cause minor heart defects [37]. These differential outcomes invite further study of how daunorubicin interacts with different genetic pathways to produce positive effects in some cell systems and adverse effects in others.

### 3.3. Identification of Functional Modules through RNA-seq

Our RNA-seq data and subsequent GO analysis indicated two groups of genes that aligned well with the chemical interactions predicted by LINCS: Wnt signaling and RNA polymerase II activity (Figure 3). As with triptolide, daunorubicin has been shown to have RNA polymerase II and III inhibitory activity [38,39,40]. It is of interest that daunorubicin has been shown to induce Jun expression in drug resistant acute myeloid leukemia cells. Loss of *gata3* caused a reduction in *junba* expression in zebrafish. Thus, the effect of these drugs may be based on restoring the normal expression of a subset of these enriched genes.

The Wnt family of ligands play an important role in the morphogenetic movements that drive formation of the pharyngeal skeleton and neurocranium. In particular, the WNT/planar cell polarity (PCP) pathway controls cell polarization and movement in these tissues [41,42,43]. The significant misregulation of several Wnt/PCP pathway members, including frizzled and frizzled-related genes, suggest that Gata3 may control effectors of the cellular movements involved in trabecular formation. In particular, *fzd7a* is required for extension of the palate in zebrafish [44]. Additionally, the *sfrp* gene family plays an important role in facilitating signal inputs in the non-canonical Wnt pathways involved in convergent extension in Xenopus [45]. Thus, genes enriched within these GO terms are excellent candidates for genes that may modify the *gata3* phenotype.

## 4. Materials and Methods

### 4.1. Animal Care and Use

Zebrafish were raised according to IACUC-approved protocols at the University of Texas at Austin and were staged as previously described [46]. All zebrafish transgenic lines were maintained from stocks derived from the AB wild-type strain. The following transgenic lines were used: *Tg(fli1:EGFP)^v1^* [47], *Tg(sox10:mRFP)^vu2^* [48], and *Tg(hsp70l;GATA3:EGFP)* [20]. These are referred to as *fli1:EGFP*, *sox10:mRFP*, and *hs:GATA3-GFP* for clarity, respectively. The *gata3^b1075^* mild and severe mutant strains were used for chemical analyses; the “mild” background was generated through an outcross to WIK and the “severe” background was generated through a cross to *fli1:EGFP*, originally in the EK background [18]. Genotyping primers for the *gata3* mutants are in Appendix A.

### 4.2. Injections and Heat Shock

The control and knockdown groups were generated by crossing *fli1:EGFP;sox10:mRFP* double transgenic fish to wild-type fish. The rescue group was generated by crossing *fli1:EGFP;sox10:mRFP* to homozygous *hs:GATA3-GFP* fish, carrying a *cmcl2:GFP* transgenesis marker. In the knockdown and rescue treatment groups, injections of a well characterized translation-blocking *gata3* morpholino 5′-CCGGACTTACTTCCATCGTTTATTT-3′ (Gene Tools, Philomath, OR, USA) were performed at the one-cell stage [49]. A 3 nL bolus of a 5 ng/μL morpholino solution was injected, a concentration that phenocopies the *gata3* mutant. Embryos were raised at 28.5 °C until 24 hpf, at which point they were placed into 15 mL conical tubes containing 1 mL of pre-heated embryo media. The tubes were submerged in 39.5 °C water for one hour to heat-shock the embryos. The embryos were then returned to fresh embryo media and incubated at 28.5 °C. This heat shock treatment has been demonstrated to rescue the facial defects in *gata3* morpholino injected embryos [20].

### 4.3. Sample Preparation and FACS

For each treatment group, 150–200 28 hpf embryos were dechorionated in a 2 mg/mL pronase solution (Sigma-Aldrich, St. Louis, MO, USA) and washed in fresh embryo medium after the chorions were removed. The embryos were briefly washed in cold calcium-free Ringer’s solution and deyolked through up and down pipetting. The deyolked embryos were collected by 300× *g* centrifugation, washed in embryo media, and placed on ice. They were then pelleted through centrifugation as before. The embryos were washed with FACSmax cell dissociation solution (Genlantis, San Diego, CA, USA) and filtered through a 40-micron cell strainer into a 35 mm Petri dish. Dissociated cells were collected in a fresh Eppendorf tube and placed on ice. Just before sorting, the cells were filtered in a polystyrene Falcon tube with a cell-strainer cap (Fisher Scientific, Hampton, NH, USA). Cell sorting was performed with a BD FACSAria Fusion SORP Cell Sorter with DIVA 8 software (BD Biosciences, San Jose, CA, USA), using a 100-micron nozzle. A 488 nm laser was used to identify GFP-expressing cells and a 561 nm laser for RFP-expressing cells. Cells were first analyzed for forward scatter and side scatter to be selected for live singlets; these cells were then sorted for fluorescence. An additional gate was added to separate *fli1:EGFP*-expressing cells from *hs:GATA3-GFP*-expressing cells (Appendix A). For each sample, approximately 100,000 cells were sorted for fluorescence. Cells from wild-type (AB) embryos of the same embryonic stage were used as a negative control. For two-color sorts, cells from embryos expressing only green or only red fluorescent markers were used as compensation controls. Double-positive cells are cranial neural crest cells and were collected into 1 mL of TRIzol (Invitrogen, Waltham, MA, USA) at 4 °C.

### 4.4. cDNA Library Preparation and RNA Sequencing

Total RNA was extracted according to the TRIzol RNA isolation protocol. Samples were purified with the RNA Clean & Concentrate kit (Zymo, Irvine, CA, USA). A Nanodrop spectrophotomer was used to determine the concentration of each sample, followed by RNA quality analysis with the Agilent BioAnalyzer (Agilent Technologies, Santa Clara, CA, USA). Total RNA from each sample ranged from 5 to 12 ng/μL. Samples were processed by the University of Texas at Austin Genomic Sequencing and Analysis Facility (GSAF). Sequencing was performed on the Illumina NextSeq 500 platform, with 75-bp paired end reads. Reads were trimmed for quality and adapters with Cutadapt v1.18 and aligned to the Genome Reference Consortium Zebrafish Build 11 (GRCz11), downloaded from Ensembl, using TopHat 2.1.1 [50,51,52]. Three biological replicates were used for RNA-sequencing.

### 4.5. Differential Expression Analysis

The dataset was checked for high read counts of *flk1*, *sox2*, and *elavl3* to ensure that there was no contamination of single-marker populations (vasculature and neurons). Normalization and gene expression analysis were performed with the R package DESeq2 [53]. An FDR-corrected *p*-value of 0.05 was used as the cut-off to identify differentially expressed transcripts. The directionally discordant subset of differentially regulated genes in the control versus knockdown and control versus rescue comparison groups comprised the final output of differentially expressed genes. The R package clusterProfiler was used to identify enriched Gene Ontology (GO) terms in gene lists using a cut-off of 0.05 FDR corrected *p*-value [54].

### 4.6. Identification and Testing of Chemical Modifiers

The Library of Integrated Network-based Cellular Signatures (LINCS) uses chemically induced transcriptomic changes in a variety of cell types to predict chemical interactions. Transcriptional changes due to a treatment (e.g., *gata3* loss of function) are compared across those within the database. Chemicals that elicit transcriptional changes that are highly positively and inversely correlated with the treatment are predicted to worsen and lessen the phenotype, respectively. The top 100 up- and down-regulated genes from the aforementioned subset of differentially regulated genes were queried against the LINCS L1000 dataset using the clue.io platform (https://clue.io/, accessed on 21 November 2022). Of the resultant nineteen chemical modifiers whose profile most resembled the transcriptomic signature, 16 (8 each with positive and negative correlation with the query) were commercially available for testing (Table 1). A dose response curve (0.1 μM, 1 μM, 10 μM, 100 μM) in wild-type embryos was used to determine the lowest observed adverse effect level (LOAEL) that elicited a craniofacial defect, including disrupted trabeculae (Appendix A). Once the LOAEL was established, a narrower range of concentrations centered on the LOAEL was used to dose *gata3^b1075^* mutant embryos. To examine gene-environment interactions, severe *gata3* mutants were treated with chemicals predicted to lessen phenotypes while mild mutants were treated with chemicals predicted to worsen the phenotypes. Embryos were treated from 24 hpf to 48 hpf to ensure potential interactions during the entire window of *gata3* expression in neurocranial progenitor cells. The embryos were then grown to 4 dpf, fixed, and stained with Alcian Blue and Alizarin Red [55]. Embryos were scored for anterior neurocranial phenotype based on the number of intact trabeculae (0, 1, or 2). Flat-mounts of stained embryos were captured using a Zeiss Axio Imager A1 microscope (Carl Zeiss AG, Jena, Germany).

### 4.7. Quantitative Reverse Transcription PCR (RT-qPCR)

We validated our RNA-seq data by testing five genes each that were up- and down-regulated from the dataset. We used Invitrogen’s SuperScript First-Strand Synthesis System for RT-PCR with oligo-d(T) primers. We performed RT-qPCR with Power Sybr Green PCR master mix (Thermo Fisher Scientific, Waltham, MA, USA, 4367659) on the Applied Biosystems QuantStudio 3 Real-Time PCR System (Thermo Fisher, A28567). QuantStudio Real-Time PCR Software was used for data analysis using the 2^–∆∆Ct^ method of relative gene expression analysis. We used the gene *csnk2b* (casein kinase 2, beta polypeptide) as an endogenous control based on its relatively unchanging expression across all experimental groups. All primers used for qPCR can be found in Appendix A. Experiments were performed in triplicate.

### 4.8. Trabecular Measurements

Linear measurements were conducted in ImageJ 1.53k. Trabecular length was measured from the boundary with the ethmoid plate to the anterior edge of the prechordal plate. Gaps between nonintact trabeculae were not included in the measurement. The midpoint of each trabecula was calculated from the length, and the width was measured at each point. Mean length and widths were calculated across control and treatment groups.

## Figures and Tables

**Figure 1 ijms-24-01222-f001:**
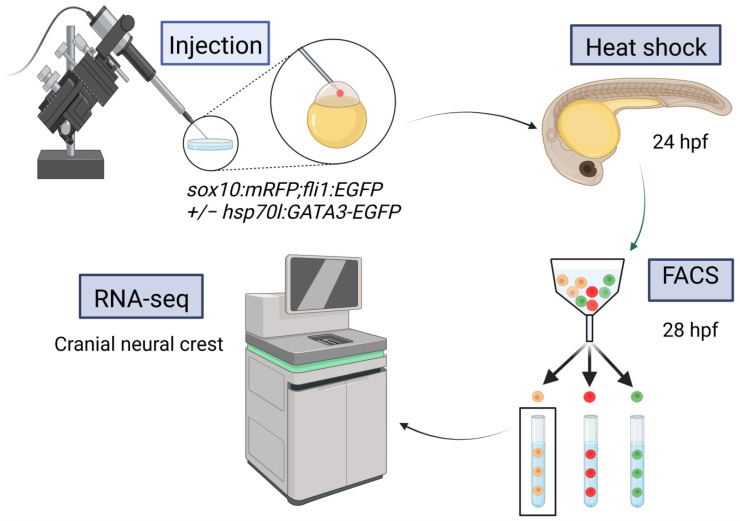
Experimental protocol for RNA-seq. Created with BioRender.com (accessed on 21 November 2022).

**Figure 2 ijms-24-01222-f002:**
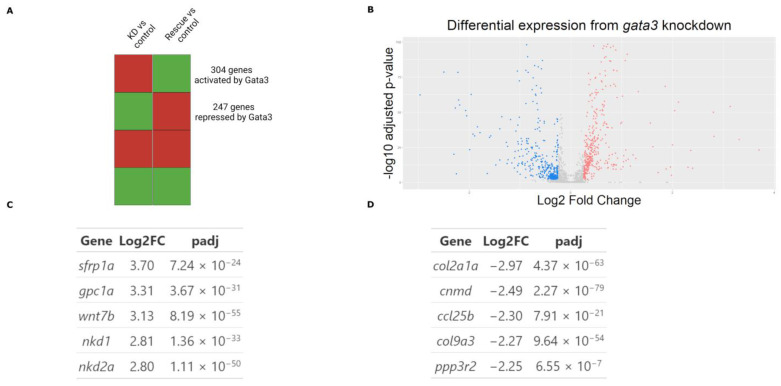
Changes in *gata3* expression drive differential regulation. (**A**) A subset of differentially expressed genes which had directionally discordant changes between the knockdown and rescue groups (relative to the control) were selected for further bioinformatic analysis. (**B**) There were 551 differentially regulated genes selected for analysis. (**C**) The five most upregulated genes in the knockdown group. (**D**) The five most down regulated genes in the knockdown group.

**Figure 3 ijms-24-01222-f003:**
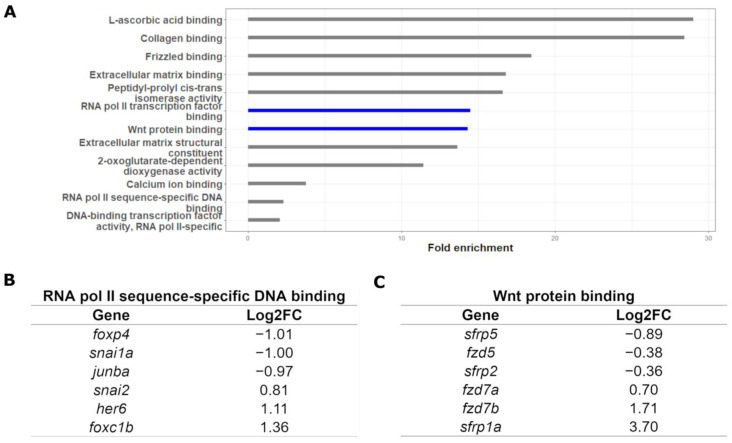
GO analysis of RNA-seq dataset. (**A**) Wnt protein binding and RNA polymerase II transcription-factor binding processes were highly overrepresented. (**B**) Representative genes from the “RNA polymerase II sequence-specific DNA binding” term. (**C**) Representative genes from the “Wnt protein binding” term.

**Figure 4 ijms-24-01222-f004:**
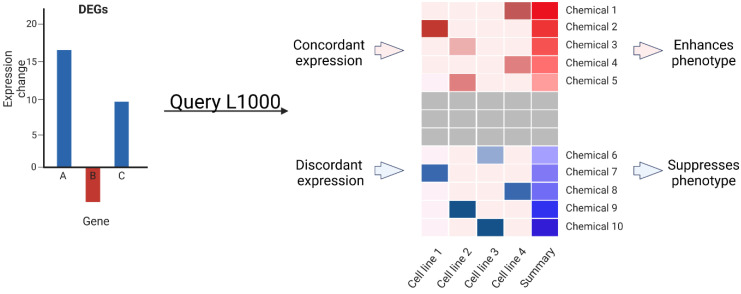
*Gata3* dataset yields chemical modifiers via LINCS. The top 100 up- and down-regulated genes affected by *gata3* knockdown and rescue were queried against the LINCS L1000 dataset. The chemical modifiers whose profile most resembled the transcriptomic signature were selected for testing. Modifiers that significantly reduced the intact trabeculae count in mild mutant phenotype are listed under “enhances phenotype”, while those that significantly increased the trabeculae count in the severe mutant phenotype are listed under “suppresses phenotype”. Created with BioRender.com.

**Figure 5 ijms-24-01222-f005:**
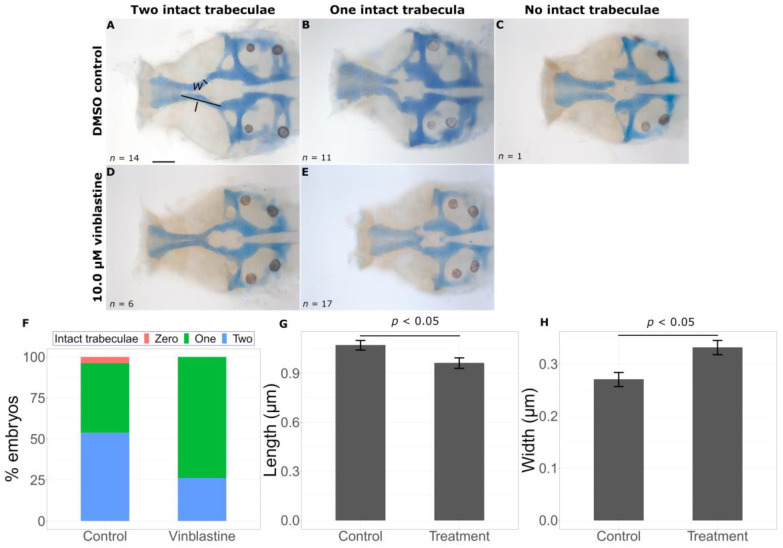
Vinblastine exacerbates the *gata3* mutant phenotype. Representative flat mounts show the range of defects in DMSO treated (**A**–**C**) and vinblastine treated (**D**,**E**) embryos. Vinblastine treatment leads to a statistically significant decrease in intact trabeculae count ((**F**), *p* < 0.05), a reduction in trabeculae length ((**G**), *p* < 0.05) and an increase in trabeculae width ((**H**), *p* < 0.05). Error bars represent SEM. Trabecular width (*w* as shown in (**A**)) measurements were taken at the midpoint of each trabecula as determined from the length (*l* as shown in (**A**)) from the boundary with the ethmoid plate to the anterior edge of the prechordal plate. Scale bar = 0.5 μm.

**Figure 6 ijms-24-01222-f006:**
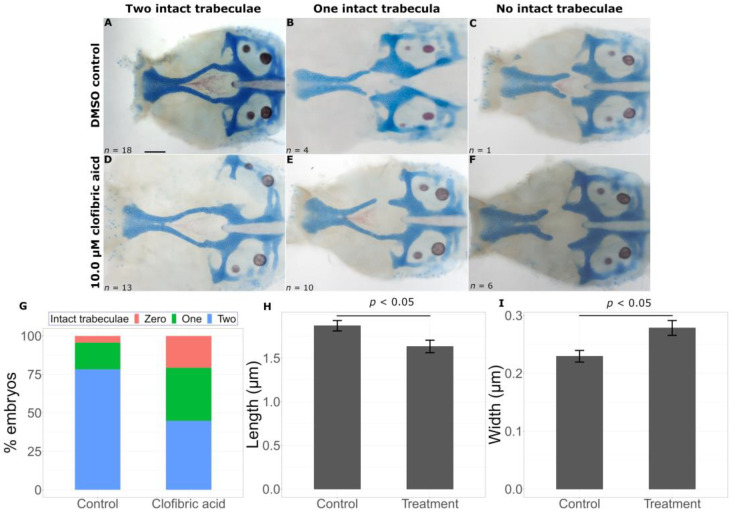
Clofibric acid worsens the *gata3* mutant phenotype. Representative flat mounts show defects in the trabeculae with DMSO treatment (**A**–**C**) and chemical treatment (**D**–**F**). Clofibric acid treatment leads to a statistically significant decrease in intact trabeculae count ((**G**), *p* < 0.05), a reduction in trabeculae length ((**H**), *p* < 0.05), and an increase in trabeculae width ((**I**), *p* < 0.05). Error bars represent SEM. Scale bar = 0.5 μm.

**Figure 7 ijms-24-01222-f007:**
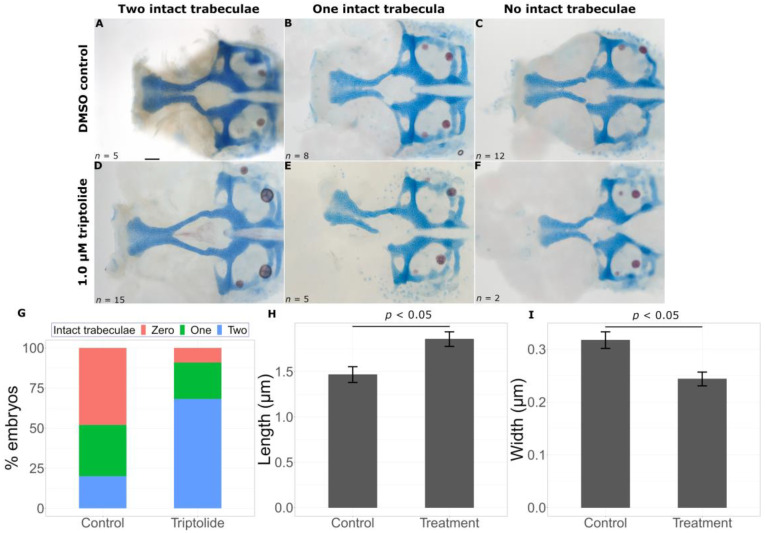
Triptolide suppresses the *gata3* mutant phenotype. Representative flat mounts show breaks in the trabeculae with DMSO treatment (**A**–**C**) and chemical treatment (**D**–**F**). Triptolide treatment leads to a statistically significant increase in intact trabeculae count ((**G**), *p* < 0.05), an increase in trabeculae length ((**H**), *p* < 0.05) and a decrease in trabeculae width ((**I**), *p* < 0.05). Error bars represent SEM. Scale bar = 0.5 μm.

**Figure 8 ijms-24-01222-f008:**
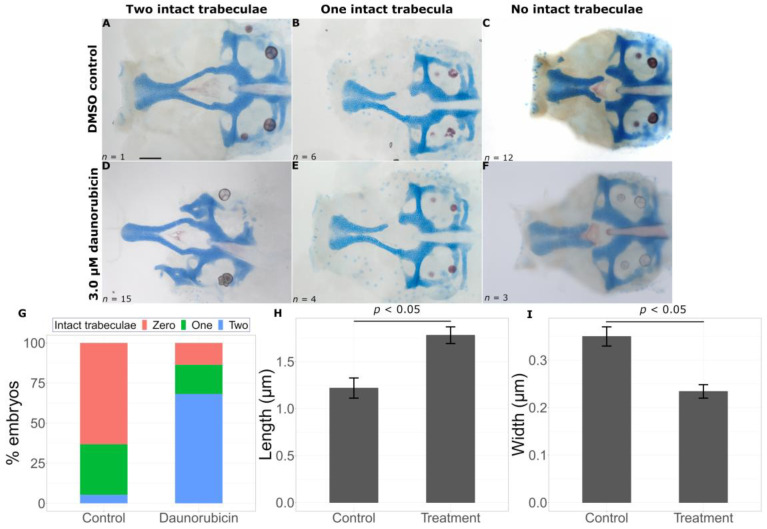
Daunorubicin suppresses the *gata3* mutant phenotype. Representative flat mounts show breaks in the trabeculae with DMSO treatment (**A**–**C**) and chemical treatment (**D**–**F**). Daunorubicin treatment leads to a statistically significant increase in intact trabeculae count ((**G**), *p* < 0.05), an increase in trabeculae length ((**H**), *p* < 0.05) and a decrease in trabeculae width ((**I**), *p* < 0.05). Error bars represent SEM. Scale bar = 0.5 μm.

**Table 1 ijms-24-01222-t001:** Predicted chemical modifiers identified by L1000 database.

Predicted Modification	Chemical	Function	CAS Registry Number
Phenotype enhancer	Flubendazole	Microtubule inhibitor, acetylcholinesterase inhibitor	31430-15-6
	Rigosertib	Cell cycle inhibitor, PLK inhibitor	1225497-78-8
	Etacrynic acid	Na/K/Cl transporter	58-54-8
	Clofibric acid	PPAR receptor agonist	882-09-7
	Nocodazole	Tubulin inhibitor, tubulin polymerization inhibitor	31430-18-9
	Vinorelbine	Tubulin inhibitor, apoptosis stimulant	71486-22-1
	Vinblastine	Microtubule inhibitor, tubulin inhibitor	865-21-4
	Vincristine	Microtubule inhibitor, tubulin inhibitor	57-22-7
Phenotype suppressor	PD-184352	MEK inhibitor, MAP kinase	212631-79-3
	Triptolide	RNA polymerase inhibitor	38748-32-2
	Belinostat	HDAC inhibitor, cell cycle inhibitor	866323-14-0
	BMS-191011	Calcium activated potassium channel opener, potassium channel activator	202821-81-6
	ISOX	HDAC inhibitor, cell cycle inhibitor	1045792-66-2
	Daunorubicin	RNA synthesis inhibitor	20830-81-3
	Chromanol	Potassium channel blocker	1481-93-2
	Alvocidib	CDK inhibitor, apoptosis stimulant	146426-40-6

## Data Availability

The data presented in this study are available in the article and the Appendix A. The raw RNA-seq data of this study will be deposited in the NCBI Sequence Read Archive prior to publication.

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
