# Peer review of "Predicting Modifiers of Genotype-Phenotype Correlations in Craniofacial Development"

_ijms, 2023, doi:10.3390/ijms24021222_

Round 1
Reviewer 1 Report
This paper is straight forward and easy to read (with an exception noted below). The conclusions are convincing and potentially clinically significant. Moreover, the paper illustrates a method that may be deployed in the other many zebrafish models of human congenital diseases to find therapeutic leads for those diseases.
My comments are all relatively minor.
1. The authors only comment on the successes. What should the reader conclude from the observation that 12 out of 18 drugs tested neither enhanced nor suppressed the phenotype, despite the predicted effects based on correlation or anticorrelation with L1000 datasets. Is this a dosing issue?
2. Page 4: ““We selected the highest concentration that yielded an embryo death rate or induced gross morphological defects in less than 10% of fish (Figure S1A).”
The analysis would be strengthened if the effects of the drugs on wild-type embryos on the number of intact trabeculae were presented. Do the drugs that enhance the phenotype of gata3 LOF embryos also reduce the number of intact trabeculae in WT embryos (i.e., is this the gross morphological defects seen in up to 10% of fish?). This would be the expectation, I think. Similarly, what do the drugs that suppress the gata3LOF phenotype do to wt embryos?
3. Page 2: “Legitimate targets of gata3 are predicted to be those whose expression moved in opposite directions when comparing the knockdown and rescue datasets (Figure 2A).
Page 2: “The resulting RNA-seq dataset yielded 701 differentially ex- pressed genes, comparing each treatment to control.
Page 3: “Figure 2. Changes in gata3 expression drive differential regulation. (A) A subset of differentially expressed genes which had directionally discordant changes between the knockdown and rescue groups (relative to the control) were selected for further bioinformatic analysis.
Text in Results, the legend to Fig 2A, and the label on Fig 2A, all imply that the rescue dataset is the differentially expressed genes (DEGs) between NC cells sorted from rescue embryos and those sorted from control embryos (e.g., the label to Fig 2A says “rescue vs control”). In Methods, control embryos are defined as uninjected wild type embryos. But the word “rescue” implies that the rescue dataset is the DEGs between NC cells sorted from gata3 morphants that either have, or have not, experienced a compensatory (rescuing) upregulation of gata3 expression from the heat-shock activated transgene. By this reasoning, the correct label for Fig 2A for the rescue dataset should be “rescue vs KD.” However, because it appears in Results this seems unlikely to be a typo. The relevance of the DEGs between rescue embryos (gata3 morphants with forced expression of gata3) and uninjected wild-type embryos needs to be explained because it seems many legitimate targets of Gata3 would be lost in this analysis. Indeed, the principle component analysis shown in Supplemental Figure S2 shows the expression profile of rescued embryos resembles that of control embryos, as expected. So of what significance is the set of DEGs between them?
4) Page 8 “To ensure that our results were truly a reflection of LINCS prediction, we randomly selected 10 chemicals from the full LINCS list, that were not correlated with the input data. We treated gata3 mutants with them as described above (Table S1). “
Page 10: “Genotyping primers for the gata3 mutants are in Supplemen- tary Table 2.
The only table in supplement is “Table S1,” which is a list of primers. There is no table listing the 10 randomly selected chemicals.
5) Page 2: “To determine the effect of gata3 on craniofacial development, we collected CNCCs from control, knockdown and rescue groups from three biological replicates and conducted RNA-seq (Figure 1). “
The reader has to refer to the Methods to learn what the “rescue” group means; this should be briefly explained in Results
6) It is unclear that how many samples in each group was processed for FACS and RNA-Seq experiments. Please clarify.
7) Fig 4A: labels defining length and width use a font that is illegibly small.
8) Page 4: “Nine of these ten genes demonstrated significant differential expression mirroring that of our RNA-seq data (Figure S2).
9) Supplemental Figure 2 shows a principle component analysis that is not referred to in the paper.
Reviewer 2 Report
The authors have developed an interesting experimental design for the prediction of chemical modifiers of the phenotypic variability that is often observed in phenotypes resulting from genomic edition in zebrafish. This design is based on the comparison of transcriptomic data obtained from zebrafish cell populations with those available in LINCS database in response to chemical compounds. Although they successfully identify four chemical modifiers of the craniofacial phenotypes, the experimental confirmation of their findings and the quality of presentation (both readability and quality of the figures) must be greatly improved to convince and to allow readers to appreciate this article.
This manuscript, in my opinion, requires major revision.
Experimental design limitations:
1) One of the main limitations of this study is the selective collection of CNCC s via FACS. As both fli1:EGFP and hsp70l:GATA3-EGFP possess green fluorescence, and cells were collected in the gata3 induced expression window, is a big concern that there could exist contamination with non CNCCs population (e.g. cells without fli1:EGFP due to GATA3-EGFP ). The authors mention that this possibility was discarded by the determination of flk1 and otx2 genes expression, although the analysis is not shown. In my opinion, due to the high bias possibility and the importance in the conclusions reached, this extent must be improved at last with an increased genetic signature analysis and a proper figure in the paper where the limitation of this approach must be discussed.
2) It is difficult to understand the rationale behind the interpretation of the “rescue” samples in the transcriptomic analysis of CNCC lines. Resulting from the simultaneous overexpression of gata3 (by hsp70l:GATA3-EGFP transgenesis) an its inhibition (by morpholinos), it is highly difficult to predict. Although it is not clearly stated in the work (and it should) it appears that the authors propose that both overpression and morpholino inhibition will neutralize each other, and the embryos will have a gata3 expression similar to control (thus the “rescue” labelling of this sample). In my opinion, the labile nature of the morpholinos makes highly improbable that the inhibition at 28 hpf will be enough to compensate the overexpression induced by the hsp70l:GATA3-EGFP transgenesis but, in any case, it must be analyzed. Western blot detection of gata3 in these populations could be a way to confirm if it is a “rescue” or a “gata3 overexpression” biological sample. Results description and discussion should be reviewed accordingly.
Quality of presentation limitations:
1) Identification of chemical modifiers of phenotype is not properly presented. Figure 3 does not show real results, only depicting a general biorender workflow scheme. It should be reworked with the data that allow to identify (chose) the chemicals shown in Table1, that should be reallocated in the 2.2 section of results, instead than discussion. Similarly, Figure 8 should be fused with Figure S3 and reallocated in the 2.1 section where those results are described.
2) Quality of figures 4, 5, 6 and 7 must be improved to allow interpretation of the results shown. Control and clorofibric bars in f Figure 5G are inverted relative to the rest of the panels and figures. Axis labelling is confusing (percent?) or misdirecting (uM can be interpreted as a concentration measure) and font size is different in each figure and even between panels in the same figure. Error bars and inserted labeling (w and l) are too small and difficult to see. The authors should rework these figures to improve presentation.
3) Several figures are miscalled in the text or are missing in the supplementary section:
- Page4, Figure S2> Missing
- Page4, Figure S1A> Figure S1
- Page8, Table S1> Missing
- Page10, Supplementary Table 2>Supplementary Table 1
Round 2
Reviewer 2 Report
Most of the limitations mentioned in the previous review have been addressed. Readability and understanding of results have been equally improved. Nevertheless, in my opinion, there are still two changes needed before publication.
1)
The figures should be numbered and presented in the order they are mentioned in the text. With the current numeration readers must leap forward and backwards to actually see the results presented in the corresponding figures. For instance, Figure 8 is mentioned and described in the second page (between Figures 2 and 3 mentions) and is allocated in the discussion section. This figure should be renumbered as Figure 3 and presented in the results section.
Also supplementary figures and tables are shuffled as they are mentioned in the text in the order:
Figure S3>Figure S4>Figure S2>Figure S1
Table S3>Table S2>Table S1
To improve readability these figures and tables should be reordered to follow the order of mention in the text.
2)
Supplementary Figure S3 represents confirmation of relative expression of DEGs measured by qPCR. These analyses (although they are normalized by a housekeeping gene) only allow to measure differences of expression in one gene between sample (morpholino) and control, but not differences of expression between different genes in the same biological sample. The figure leads to the interpretation that in control samples there are differences in expression between genes (for instance, nkd2a is overexpressed in controls relative to nkd1 gene). This represtantion is misleading and the expression of each gene should be additionally normalized relative to the control biological sample (all genes should have a mean expression of 1 in the control sample).
Congratulations for a very interesting study. I’m sure it will be valued by researchers in the field of developmental genetics.
